# CircSRSF4 Enhances Proliferation, Invasion, and Migration to Promote the Progression of Osteosarcoma via Rac1

**DOI:** 10.3390/ijms23116200

**Published:** 2022-05-31

**Authors:** Xiaoya Sun, Xuefeng Zhao, Shuangyan Xu, Yang Zhou, Zhen Jia, Yuebai Li

**Affiliations:** Department of Biochemistry and Molecular Biology, School of Basic Medical Sciences, Zhengzhou University, Zhengzhou 450001, China; 18790690160@163.com (X.S.); zhaoxuefeng717@163.com (X.Z.); xsyblb123@163.com (S.X.); 15138638288@163.com (Y.Z.); jiaz0104@sina.com (Z.J.)

**Keywords:** circSRSF4, miR-224, Rac1, osteosarcoma, proliferation

## Abstract

(1) Background: At present, cancer cell metastasis is the main cause of death in patients with malignant tumors, and up to 23% of osteosarcoma patients have died due to lung and lymph node metastasis. Therefore, finding new molecules involved in tumor development can provide new strategies for the diagnosis and treatment of osteosarcoma patients. Circular RNAs (circRNAs) are a type of RNA molecule that are connected head-to-tail to form a closed ring. There is increasing evidence that circRNAs are RNA molecules with many biological functions in various diseases. However, the role and mechanism of circRNAs in osteosarcoma have rarely been reported. (2) Methods: The expression of circSRSF4 in osteosarcoma tissues and cell lines was detected by quantitative real-time PCR (RT-qPCR), and the result of high-throughput sequencing was verified. In order to explore the effect of circSRSF4 on tumor proliferation, invasion, and migration, a dual-luciferase reporter assay, RNA binding protein immunoprecipitation assay, cell counting kit-8 (CCK-8), transwell assay, scratch wound healing assay, Western blot analysis, and other experiments were carried out in vitro. Rescue experiments and a xenograft model confirmed that circSRSF4 directly acted on miR-224 to regulate Rac1 expression. (3) Results: The expression of circSRSF4 was significantly higher in osteosarcoma tissues and cell lines. Down-regulating the expression of circSRSF4 in vitro significantly inhibited the proliferation, invasion, and migration of cells, and also reduced the expression of Rac1, while the overexpression of Rac1 and miR-224 inhibition could reverse these effects. The inhibition of circSRSF4 expression in vivo also attenuated tumor growth. A mechanistic study showed that circSRSF4 can be used as an miR-224 sponge to up-regulate the expression of Rac1, thereby promoting the development of osteosarcoma. (4) Conclusions: CircSRSF4 acting as a ceRNA promotes the malignant behavior of osteosarcoma through the circSRSF4/miR-224/Rac1 axis, which provides a new theoretical basis for the clinical prevention and treatment of osteosarcoma and the study of related markers and intervention targets.

## 1. Introduction

Osteosarcoma (OS) is one of the most common malignant bone tumors originating from osteoblasts [1]. It can occur at any age; the incidence in children and adolescents is high, but it also increases after 60 years of age [2]. The five-year survival rate of osteosarcoma patients is only 60% to 70%, and the cells are prone to metastasize to the lung [3], while patients diagnosed with cancer cell metastasis or recurrence have a survival rate of less than 30% within 5 years [4]. At present, it has been found that the occurrence, development, and metastasis of osteosarcoma involve the abnormal regulation of various biomolecules, such as the imbalance between oncogenes and tumor suppressor genes [5,6], the participation of multiple cytokines, the imbalance between cell proliferation and apoptosis [7], the under-expression or over-expression of genes [8,9], or gene mutation [10], and many other factors contribute to the occurrence of osteosarcoma. Surgery combined with chemotherapy is currently the most effective treatment for osteosarcoma, but the existing treatment has a poor therapeutic effect on osteosarcoma patients with metastasis and recurrence. Therefore, finding new molecules involved in tumor development can provide new strategies for the diagnosis and treatment of osteosarcoma patients, and targeted therapy shows great potential in the treatment of osteosarcoma, but effective therapeutic targets still need to be explored.

Circular RNA (circRNA) is a kind of special RNA molecule that is spliced end-to-end and does not have 3′ and 5′ ends [11,12]. The lack of free ends of circRNA makes it resistant to the cleavage of the exonuclease and, therefore, it has stable expression in the cytoplasm [13]. When scientists first discovered circRNA, they tended to think that circular RNA was just a by-product of the alternative splicing of a gene and had no practical function; they even called it “dark matter” or “junk sequence” [14,15]. However, recent studies have found that they play an important role in the development of cancer and they have become a current research hotspot. One of the functions of circRNA is to combine with specific miRNAs as a competitive endogenous RNA in tumors to relieve the inhibition of miRNAs on the target protein mRNA, affect the gene-mediated cell signal transduction, and then regulate the progress of various malignant tumors. [16,17,18]. MicroRNA (miRNA) is a type of short single-stranded RNA (usually 21–22 nucleotides) without a coding function, which is widely expressed in the organism. Many studies have found that miRNAs can regulate a large number of tumor-related genes and play a role in inhibiting or promoting cancer [19]. At present, the research of gene therapy and non-coding RNA (ncRNA) has made a breakthrough, and the function of non-coding RNA is also being revealed. The in-depth study of non-coding RNA not only enriches the understanding of gene regulatory networks in theory, but it is also a potential new biological marker and therapeutic target for disease diagnosis in clinical application.

Ras-related C3 botulinum toxin substrate 1(Rac1) is a member of the Rho family of the small GTPase superfamily, with a molecular size of 29 kb and is located on the human chromosome 7p22. The study of Rac1 mainly focuses on its role in cell adhesion, cell migration, and cytoskeleton formation [20]. Rac1 is widely expressed, especially in malignant tumor tissues. It has been found that the dysregulated expression of GEFs, which are responsible for Rac1 activation, is closely related to tumor metastasis and drug resistance [21]. Rac1 can affect the expression of cyclin and then regulate the proliferation of cancer cells. If the expression of Rac1 is inhibited, the expression of Cyclin D1 can be down-regulated, and the cell cycle is blocked in the G1/S phase [22]. It can also promote the division and proliferation of tumor cells by negatively regulating the expression of p21 [23]. Under the action of PI3K (phosphatidylinositol 3 phosphate), Rac1-GDP can be converted into Rac1-GTPase, thereby activating tumor metastasis-related transduction pathways in which Rac1 is involved [24]. In short, Rac1 can participate in different signal pathways to mediate the occurrence and development of malignant tumors.

In this study, we first demonstrated that circSRSF4 (serine/arginine-rich splicing factor 4), which was found by high-throughput sequencing and validation screening, is indeed present in osteosarcoma cells, and highly expressed in osteosarcoma tissues and cell lines. Secondly, we observed that Rac1 promotes the proliferation, growth, migration, and invasion of osteosarcoma cells. Further in vitro and in vivo mechanism studies showed that circSRSF4 can down-regulate the expression of Rac1 through binding with miR-224 competitively, which eventually leads to the weakening of proliferation, invasion, and migration of osteosarcoma cells. Therefore, our results confirm that circSRSF4 plays a key role in the development and metastasis of osteosarcoma by regulating the function of Rac1, and may become a new target and strategy for clinical prevention and treatment of osteosarcoma.

## 2. Materials and Methods

### 2.1. Patients and Clinical Specimens

In this study, all osteosarcoma tissues and matched paracancerous tissues were collected from the First Affiliated Hospital of Zhengzhou University. A total of 13 pairs of specimens from patients whose pathological type was identified as osteosarcoma by pathological biopsy of the First Affiliated Hospital of Zhengzhou University were selected. All participants provided written informed consent. The pathological characteristics of patients are shown in Table 1. Fresh tumor tissues were washed, labeled, and quickly frozen in liquid nitrogen for subsequent q-PCR, Western blot analysis, and other experiments. This study complies with all relevant ethical standards and is approved by the Ethics Committee of Zhengzhou University. All procedures are conducted in accordance with the principles of the Helsinki Declaration.

### 2.2. Cell Lines and Cell Culture

The human osteosarcoma cell lines MG63, SAOS-2, U2OS, and 293T used in this experiment were purchased from the Cell Resource Center of Shanghai Institutes for Biological Sciences, and the normal human osteoblast cell line hFOB1.19 was purchased from Shanghai Kuisai Biotechnology Co., Ltd. The hFOB 1.19 cell line was cultured in DMEM/F12 (VivaCell, Shanghai, China),0.3 mg/mL G418, and 10% FBS (Gibco, Waltham, MA, USA) and maintained at 33.5 °C in 5% CO_2_, whereas all other lines (293T, U2OS, MG63, SAOS-2) were cultured in DMEM (VivaCell, Shanghai, China) containing 10% FBS (VivaCell, Shanghai, China) and 1% penicillin/streptomycin at 37 °C in 5% CO_2_.

### 2.3. RNA Isolation and Quantitative Real-Time PCR

The total RNA from the tissue samples and cells was extracted according to the introduction of Trizol reagent. The cDNA synthesis and q-PCR of circSRSF4 and Rac1 were conducted according to the instructions of the Takara RNA PCR kit and SYBR Premix Ex Taq II (TliRNaseH Plus), respectively, while the cDNA synthesis and q-PCR of miR-224 were in accordance with the Hairpin-itTM microRNA and U6 snRNA Normalization RT-PCR Quantitation Kit and SYBR Green Master Mix (Vazyme, Nanjing, China) instructions, respectively. Finally, analysis was performed on an Applied Biosystems 7500 Real-time PCR System (Thermo Fisher, Waltham, MA, USA). The primer sequences are shown in Table 2. 

### 2.4. Identification of CircSRSF4

The obtained cDNA and gDNA were used as templates, and divergent primer and convergent primer as primers, respectively, for PCR amplification. Then, the DNA fragments recovered from agarose gel were used for T-vector attachment (4 °C, 12 h). After the transformation in competent DH5α *E. coli*, blue-white screening was performed, and white colonies were cultured for sequencing analysis.

### 2.5. Cell Transfection

The cells were seeded at a density of 3 × 10^5^ cells/well and transfected when the cells reached 50%. The circSRSF4 siRNA, circSRSF4 NC sequence, miR-224 mimics, miR-224 inhibitor, miR-224 NC sequence, and pcDNA3.1-Rac1 plasmid was added to 100 μL of Opti-MEM with enzyme-free pipette tips and left to stand for 5 min. At the same time, 15 μL of LipofectamineTM 2000 was added to 100 μL of pre-chilled Opti-MEM, also left standing for 5 min. Then, the two liquids were mixed and left for 20 min. The cells were washed twice with PBS, and the transfection mixture was slowly added. After being placed in the incubator for 5 h, the cells were replaced with 2 mL of new medium. The culture was continued for another 48 h, and the transfection efficiency was detected by RT-qPCR.

### 2.6. Western Blot Analysis

The total protein concentration was quantified using the Bicinchoninic Acid (BCA) kit after isolating the total protein from the cells with radioimmunoprecipitation (RIPA) buffer containing phenylmethanesulfonyl fluoride (PMSF). After SDS-PAGE electrophoresis, it was transferred to a PVDF membrane and blocked with 5% skimmed milk for 2 h. The PVDF membrane was washed three times with Tris-buffered saline containing Tween-20 (TBST) and incubated with primary antibodies against Rac1 and β-actin. After washing the membrane three times with TBST, HRP-conjugated anti-rabbit and anti-mouse secondary antibodies were used. A signal was detected using ECL Western blotting substrate.

### 2.7. Cell Counting Kit-8 (CCK-8) Cell Viability Assay

Cells were cultured in a 96-well plate at a density of 5 × 10^3^ cells/mL. Five duplicate wells were set for each group, and PBS was added around the experimental wells to avoid the evaporation of the medium. After 24 h incubation, 10 μL of CCK-8 solution was added to each well (obliquely adhered to the wall to avoid air bubbles and affecting the absorbance measurement), and the culture plate was gently shaken. Then it reacted at 37 °C for 1 h, and the absorbance was detected at 450 nm with a microplate reader. CCK-8 solution (10 μL) was added to each well every 24 h for reaction and then tested again. We collected the data and drew the growth curves.

### 2.8. Transwell Invasion Assay

We diluted the liquid BD Matrigel with pre-chilled DMEM (ratio of 8:1) and added it to the chamber of a Transwell Permeable Support device. It was then placed in a 37 °C incubator and dried for 5 h. Then, 200 μL of cells with a density of 2 × 10^5^ cells/mL were added to the upper chamber, and 700 μL of DMEM containing 10% FBS medium to the lower chamber. Each group was provided with 5 duplicate wells and were cultured for 24 or 48 h. After gently wiping the upper chamber with a cotton swab, we placed the chamber in 4% paraformaldehyde for 15 min, washed it three times with PBS, and then stained it with DAPI for 5 min. The cells on the chamber were washed three times with PBS and placed in a new culture plate. Different fields of view were randomly selected, and the cells were counted under a 200× magnification. The picture was saved and the data was analyzed.

### 2.9. Scratch Wound Healing Assay

We added 3 × 10^4^ cells/well to the six-well plate and continued culturing to 80–90% confluence. The cell layer was scratched with the tip of a sterile 10 μL pipette tip, and the damaged cells were carefully washed away with PBS and then cultured with serum-free DMEM. After 0 h, 24 h, and 48 h of incubation, the pictures were taken under a 100× magnification.

### 2.10. RNA Binding Protein Immunoprecipitation (RIP)

According to the protocol of EZ-Magna RIP kit (Millipore, Burlington, MA, USA), after collecting 10^7^ cells, RIP lysis buffer containing protease inhibitor cocktail and RNase inhibitor was added, followed by 40 min on ice and centrifuged at 4 °C for 15 min. (12,000× *g*). Then incubate 4 h at 4 °C with RIP buffer containing magnetic beads bound to anti-argonaute 2 (Ago2) antibodies and negative control IgG. After RNA extraction from the samples, the enrichment of circSRSF4 was identified by q-PCR. The whole experiment should be performed under the condition of RNase free.

### 2.11. Immunohistochemistry (IHC)

The tissue specimens were fixed in 4% paraformaldehyde, then embedded in paraffin and sectioned. The tissue sections were deparaffinized in xylene, rehydrated with gradient alcohol, and boiled in sodium citrate for 30 min for antigen retrieval. Then, the tissue sections were blocked in 10% normal goat serum for 15 min and incubated with the primary antibodies against Ki-67 overnight at 4 °C. We washed the tissue sections in PBST and incubated them with the secondary antibody for 2 h at room temperature. A Vectastain Elite DAB kit (Vector Laboratories, Burlingame, CA, USA) was used to detect immunoreactivity.

### 2.12. Bioinformatics Analysis

CircBase (http://www.circbase.org/ accessed on 11 April 2022) and Circular RNA Interactome, bioinformatics software, were used to screen and analyze the possible microRNA that circSRSF4 may interact with. TargetScan (www.targetscan.org/ accessed on 11 April 2022)), miRBase (www.mirbase.org/ accessed on 11 April 2022)), and other software analyzed and predicted the target gene Rac1, which microRNA might target. The screening results were analyzed to select the microRNA that not only bound to the 3′-UTR of Rac1 mRNA but also interacted with circSRSF4. Finally, miR-224 was selected as the research object, and the bioinformatics software was used to predict the possible sites of miR-224 affecting the circSRSF4 and Rac1, respectively.

### 2.13. Dual-Luciferase Reporter Gene Assay

We co-transfected the miR-224 mimics and the miR-224 negative control and successfully constructed recombinant vectors pmirGLO-circ and pmirGLO-Rac1 (Wild-type/Mutant-type) into cells with LipofectamineTM 2000. After 48 h of incubation, the luciferase activity was measured using the dual-luciferase reporter system (Promega) according to the manufacturer’s protocol. Combined with the fluorescence of the blank control group, the relative activity of the firefly luciferase was determined.

### 2.14. Establishment of Stable Cells

A mixture of puromycin and medium with a concentration of 0–10 µg/mL was added into MG63 cells in gradients and observed for 5 days to obtain the minimum concentration of puromycin that can kill the cells. The cells were infected by a diluted titer of 1/10^8^ of circSRSF4 siRNA lentivirus and circSRSF4 negative control lentiviral drops (Genema). After culturing for 24 h, the medium was replaced with fresh medium, and the optimal concentration of puromycin was added after 48 h to screen the cells. After 72 h, the surviving infected cells were collected and used for a tumor formation experiment in nude mice.

### 2.15. Xenograft Model

All animal studies were approved by the Animal Research Ethics Committee of Zhengzhou University. BALB/C female nude mice (5 weeks old) lived in a specific pathogen-free (SPF) facility and were given SPF feed. The underarms on the left and right sides of the nude mice were injected with 0.1 mL of stable cells of si-circSRSF4 and si-circSRSF4 NC with a concentration of 1 × 10^7^ cells/mL and observed every 2 days. Tumors began to appear in the armpit of nude mice at about 5 days, and the tumor sizes were measured and recorded every 5 days thereafter. After 20 days, the nude mice were sacrificed, and the tumor tissues were removed and weighed. Some fresh tissue was put into an EP tube and 1 mL of pre-chilled tissue fixation solution was added for immunohistochemical analysis. The remaining tissue was used for qRT-PCR and Western blotting.

### 2.16. Statistical Analysis

The data were presented as the mean ± standard deviation (SD) and all statistical analyses were performed using SPSS 20.0 software (IBM Corp., Armonk, NY, USA). Normal analysis and homogeneity of variance tests were conducted first. Differences between the two groups were compared using Student’s *t*-test, and differences between multiple groups were compared using one-way ANOVA. The non-parametric Spearman rank correlation analysis was used to calculate the correlation coefficient, and the chi-square test was used for comparison of the classification data. A Kaplan–Meier survival curve was used to analyze the overall survival data of the patients. A *p*-value of < 0.05 was used as a criterion to evaluate significant differences among the statistics.

## 3. Results

### 3.1. Identification of hsa_circ_0006602 and Its Expression in Tissues and Cells

From the results of high-throughput sequencing, it was found that hsa_circ_0006602 was significantly differentially expressed in osteosarcoma tissues and its adjacent tissues (log2FC = 5.4, *p* = 1.83 × 10^−10^), so hsa_circ_0006602 was selected as the research object. The expression of hsa_circ_0006602 in osteosarcoma tissues and osteosarcoma cell lines were significantly higher than that in the corresponding paracancerous tissues and the human normal osteoblast cell line hFOB1.19 (Figure 1A,B). Hsa_circ_0006602 originated from the 3’ AG sequence and the 5’ GA sequence of the exon 4 of the recombinant serine/arginine-rich splicing factor 4 (SRSF4) gene, which is located at chromosome 1 (Figure 1C). The base sequence of the cleavage site is different from that of SRSF4. We designed a divergent primer with the splice site and a convergent primer without the splice site, and a PCR and agarose gel electrophoresis assay were performed using the cDNA and gDNA of the osteosarcoma cells as the templates, respectively. The results show that hsa_circ_0006602 could be detected by the divergent primer in cDNA, but the band detected by the convergent primer was different. Meanwhile, the band was not detected by the divergent primer in gDNA (Figure 1D, M: marker), proving that the circRNA was reversely cyclized. The divergent primer was used for PCR amplification, and the product was recovered and cloned into a T vector for Sanger sequencing. The results show that a back-splice junction was formed between the 3′ AG sequence and the 5′ GA sequence of exon 4 of the SRSF4 gene (Figure 1E). Therefore, hsa_circ_0006602 was a circRNA formed by the reverse splicing of the exon 4 of the SRSF4 pre-mRNA. We refer to hsa_circ_0006602 as circSRSF4 hereafter.

### 3.2. CircSRSF4 Is Required for Promoting the Proliferation, Migration, and Invasion of Osteosarcoma Cells

To clarify the role of circSRSF4 in osteosarcoma cells, we selected the MG63 and U2OS cell lines as the research object. The siRNA was also constructed against circSRSF4 (Figure 2A). The siRNA of circSRSF4 was transfected into cells by liposomes, and the transfection efficiency was verified. Compared to the blank group (cells without any treatment), the expression of circSRSF4 in the si-NC group was not significantly changed. The expression of circSRSF4 in cells infected with si-circ was significantly decreased (Figure 2B), indicating that siRNA to circSRSF4 was successfully transfected into the cells.

The results of the CCK-8 show that compared to the blank group (cells with transfection reagent only) and negative control group, the proliferation ability of the cells in the circSRSF4 knock-down group (si-circ) was significantly inhibited (Figure 2C). The transwell assay showed that the invasive ability of cells in si-circ group was significantly reduced (Figure 2D). The results of the cell scratch wound healing assay show that the migration ability of the cells in the si-circ group was significantly inhibited (Figure 2E). Therefore, these show that the proliferation, invasion, and migration of osteosarcoma cells can be inhibited after down-regulating the expression of circSRSF4.

### 3.3. CircSRSF4 Binds to miR-224 Directly and Down-Regulates Its Expression

As a competitive endogenous RNA (ceRNA), circular RNA can combine with specific miRNAs and regulate the expression of miRNAs as a miRNA sponge [25,26]. According to the circular RNA Interactome database, we predicted the miRNAs that circSRSF4 might sponge bind to and finally screened miR-224 (Figure 3A). Then, we detected the expression of miR-224 in tissues and cells. The results of the qRT-PCR show that the expression of miR-224 in osteosarcoma tissues was significantly lower than that in the adjacent tissues (Figure 3B), and the expression in the three osteosarcoma cell lines was significantly lower than that of normal human osteoblasts (Figure 3C). 

We knocked down circSRSF4 (si-circ) and found that the expression of miR-224 increased after the down-regulation of circSRSF4 (Figure 3D), indicating that down-regulation of circSRSF4 can promote the expression of miR-224. In order to further confirm the direct interaction between circSRSF4 and miR-224, we used a luciferase reporter assay for verification. The design of the pmirGLO-circSRSF4 wild/mutant type vector is shown in Figure 3E. The results show that in human 293T cells, the relative activity of luciferase was significantly reduced by the co-transfection of wild-type pmirGLO-circSRSF4 and miR-224 mimics (Figure 3F). However, there was no significant change in the relative activity of luciferase between the co-transfected miR-224 mimics and mutant pmirGLO-circSRSF4 group and the co-transfected miR-224-NC and wild-type pmirGLO-circSRSF4 group. Moreover, as verified by the assays, the results of the RNA immunoprecipitation assay indicate that circSRSF4 could target miR-224 in a direct manner (Figure 3G). Instructively, circSRSF4 can bind to miR-224, thereby down-regulating its expression.

### 3.4. CircSRSF4 Served as a ceRNA for miR-224 to Regulate the Expression of Rac1

Many researchers have reported that one of the functions of miRNA is to specifically bind to the target mRNA and affect the protein synthesis process, which mainly plays a negative role in regulating gene expression [27]. According to previous studies, Rac1 participates in the tumor metastasis-related transduction pathways and is closely related to the occurrence and development of osteosarcoma [24,28]. Therefore, we chose Rac1 as the target gene for further research. We detected the expression of Rac1 in tissues and cells. The results of the qRT-PCR and Western blot analysis show that the expression of Rac1 mRNA and protein in osteosarcoma tissues were significantly higher than those in the adjacent tissues (Figure 4A,B, N: normal adjacent normal tissues, T: tumor osteosarcoma tissue). This is consistent with the detection results in the three osteosarcoma cell lines and normal osteoblasts cells (Figure 4C,D). The expression of miR-224 in the OS cell lines was relatively low (Figure 3C), while the expression of Rac1 was relatively high, so it was speculated that there may be targeted regulation between miR-224 and Rac1. Therefore, after the overexpression of miR-224, we used qRT-PCR and Western blotting to detect the transfection efficiency and expression of Rac1. Compared to the blank group (cells with transfection reagents only), there was no significant change in the expression of miR-224 after transfection with miR-224-NC, while it increased significantly after transfection with the miR-224 mimics (Figure 4E). These results show that miR-224 mimics were successfully transfected into MG63 cells. Compared to the blank group and the negative control group, the expression of Rac1 mRNA and protein in the miR-224 mimic group was significantly down-regulated (Figure 4F,G). The results indicate that miR-224 can down-regulate the expression of Rac1. In order to further confirm the direct interaction between miR-224 and Rac1, we used a luciferase reporter assay for verification. The design of the pmirGLO-Rac1 wild/mutant type vector is shown in Figure 4H. The results show that the relative activity of luciferase was significantly reduced after co-transfection of the miR-224 mimics and wild-type pmirGLO-Rac1 in 293T cells (Figure 4I). However, there was no significant change in the relative activity of luciferase between the group co-transfected with miR-224 mimics and mutant pmirGLO-Rac1 and the group co-transfected with miR-224-NC and wild-type pmirGLO-Rac1. It can be concluded that miR-224 targets the 3′UTR region of Rac1 mRNA and negatively regulates its expression. Interestingly, we found that after down-regulating the expression of circSRSF4 (si-circ), the expressions of Rac1 mRNA and protein were reduced (Figure 4J,K). To summarize, these data indicate that circSRSF4 acted as a sponge for miR-224, thereby regulating the expression of Rac1 in osteosarcoma.

### 3.5. CircSRSF4 Promotes OS Cell Proliferation, Migration, and Invasion through the circSRSF4/miR-224/Rac1 Axis

In order to verify whether circSRSF4 promoted the proliferation, invasion, and migration of osteosarcoma through the circSRSF4/miR-224/Rac1 axis, we co-transfected the Rac1 overexpression plasmid with the 3’UTR region and circSRSF4 siRNA into osteosarcoma cells for a rescue experiment. The results of the CCK8, transwell, and scratch wound healing assays show that the overexpression of Rac1 could reverse the inhibitory ability of circSRSF4 siRNA for proliferation, invasion, and migration in cells (Figure 5A–C). In addition, after the co-transfection of the circSRSF4 siRNA and miR-224 inhibitor, the results of the qRT-PCR and Western blotting also show that the down-regulation of Rac1 caused by knocking down the circSRSF4 could be partially recovered by inhibiting the expression of miR-224 (Figure 5D,E). Collectively, these results reveal that the progression and development of OS depend on the circSRSF4/miR-224/Rac1 axis.

### 3.6. Down-Regulation of CircSRSF4 Inhibits the Growth of OS In Vivo

To further explore the effect of circSRSF4 on the proliferation of osteosarcoma cells in vivo, cells stably transfected with circSRSF4 siRNA and a circSRSF4 negative control were inoculated subcutaneously in BALB/c nude mice. The tumor volumes were measured every 7 days, and the tumor weights were measured after 20 days; then, the tumor growth curve was drawn. Compared to the control group, the volumes, growth, and weights of the tumors in the circSRSF4 siRNA group were significantly reduced (Figure 6A), and immunohistochemical staining analysis showed that the expression of Ki-67 was lower than that in the control group (Figure 6B). The expression of circSRSF4, miR-224, and Rac1 in the subcutaneous tumorigenic tissues of the nude mice were detected by qRT-PCR and Western blot analysis. The results show that in osteosarcoma tissues inoculated with cells stably transfected with circSRSF4 siRNA, the expression of circSRSF4 was significantly lower than that of the control group (Figure 6C), the expression of miR-224 was significantly higher than that of the control group (Figure 6D), and the expression of Rac1 was significantly lower than that of the control group (Figure 6E,F). To sum up, after knocking down circSRSF4, the expression of circSRSF4 and Rac1 were down-regulated and the expression of miR-224 was up-regulated in the tumorigenic tissues of the nude mice. These findings demonstrate that circSRSF4, as an miR-224 sponge, promoted the progression of osteosarcoma by increasing the expression of Rac1.

## 4. Discussion

CircRNA (circular RNA) is a new member of the non-coding RNA family with great potential. Its unique circular structure givescircRNA a variety of biological characteristics that are different from those of previous linear RNA. Nigro et al. discovered transcripts produced by non-canonical splicing caused by deletion in the colon cancer gene [29]. Later, Cocquerelle proved the circularization of these transcripts, but he called these molecules ‘mis-spliced’ [30]. Although the function of circRNAs has been discovered gradually with the deepening of research, there is still the opinion that they could be a by-product of splicing. In 2014, it was demonstrated that circRNAs are produced co-transcriptionally, and, moreover, that they could compete with conventional splicing [31]. Therefore, the production of circRNA results in decreased mRNA at the same locus. CircRNA is produced by spliceosomes, however, there are many circular RNAs formed by exons that are not produced by alternative splicing [32,33]. Thus, some enrichment of circRNAs could regulate the production of host gene mRNAs [34]. In addition to alternative splicing, the production of circRNAs is also associated with inefficient cleavage and polyadenylation [35].CircRNA is rich in species, stable in structure, conserved in sequence, and can be specifically expressed in various organism tissues or cells [34]. Especially in the study of human tumor diseases, circRNA participates in various signal regulation networks. For example, in various malignant tumors (such as lung cancer, cervical cancer, neuroblastoma, and so on), the high expression of ciRS-7 is closely related to tumor progression [36]. In this study, we detected a high expression of circSRSF4 in both osteosarcoma tumor tissues and cells, which is consistent with our high-throughput sequencing results of osteosarcoma tissues, and we speculated that circSRSF4 may play an important role in the initial process of osteosarcoma. We first identified the structure of circSRSF4 in osteosarcoma. After the down-regulation of circSRSF4 by RNA interference, we found that the proliferation, invasion, and migration abilities of osteosarcoma cells were reduced. In nude mice tumorigenesis experiments, the tumor growth of MG63 and U2OS cells with transfected siRNA of circSRSF4was significantly inhibited. This shows that circSRSF4 has a significant effect on the malignant behavior of osteosarcoma cells, and one of the important functions of circRNA is that it can act as a miRNA sponge to specifically adsorb miRNA and then regulate gene expression. The functions of circRNA that have been reported so far are mainly this type of action.

Rajewsky, a pioneer in circRNA research, and his colleagues found that circRNA CDR1as has more than 60 binding sites with miRNA-7 [37]. The Ge Feng team found that CDR1as affects the expression of the epidermal growth factor by “adsorbing” miRNA-7, which ultimately affects the cell proliferation of liver cancer cells [38]. More and more studies have shown that the ceRNA mechanism is the main way for circRNAs to exert their biological functions [25,39,40]. In gastric cancer, for example, circCCDC9 exerts a tumor-suppressive effect through the miR-6792-3p/CAV1 axis and is significantly related to the tumor size, infiltration of the lymph node, and overall survival of GC patients [39]. Moreover, it has been reported that has-circRNA-5692 can be used as a sponge of miR-328-5p, regulating the expression of the downstream gene DAB2IP and inhibiting the progression of liver cancer [41], but there are few reports about circRNA in osteosarcoma. In this study, the binding sites of circSRSF4 and miR-224 were speculated and verified through bioinformatics analysis and a luciferase reporter assay. We also confirmed that miR-224 mimics/miR-224 inhibitors can reverse the up-regulation/down-regulation of Rac1 expression caused by circSRSF4/circSRSF4 si-circ in osteosarcoma cells. These results fully prove that circSRSF4 is a miR-224 sponge and also suggest that miR-224 is a key negative regulator of Rac1. There is a possibility that there is a network regulation relationship between circSRSF4, miR-224, and Rac1 in osteosarcoma. The research of Rac1 mainly revolves around its role in cell adhesion, cell migration, and cytoskeleton formation, and it belongs to the Rho family of the small GTPase superfamily [20]. Rac1 is highly expressed in esophageal squamous cell carcinoma (ESCC) and is closely related to poorer prognosis in ESCC patients as well as enhanced RAC1 expression and increased cell proliferation, migration, and chemoresistance [42]. More studies have found that abnormal expression of Rac1 in thyroid cancer and liver cancer can regulate the malignant behavior of tumors [43,44]. In non-small cell lung cancer, Rac1 is significantly up-regulated in tissues, while the knock-down of Rac1 can inhibit the invasion and migration of tumor cells and improve the sensitivity of cells to anticancer drugs [45,46]. Our results show that Rac1 mRNA and protein were highly expressed in osteosarcoma tumor tissues and cells, suggesting that Rac1 is closely related to the occurrence and malignant biological characteristics of osteosarcoma. In this study, we predicted and confirmed that miR-224 can target the Rac1 3’-UTR region through bioinformatics analysis and a luciferase reporter assay and negatively regulate the expression of Rac1. In the rescue experiments, the overexpression of Rac1 partially restored the promotion of the proliferation, migration, and invasion abilities of MG63 and U2OS cells caused by circSRSF4. The above results further verify that circSRSF4 served as a ceRNA for miR-224 and promoted the expression of Rac1.

In summary, our research revealed the key role of the circSRSF4/miR-224/Rac1 axis in the development of osteosarcoma. This is the first in-depth study of the mechanism of circSRSF4 in osteosarcoma, which provides a new theoretical basis for the clinical prevention and treatment of osteosarcoma and the research of tumor-related markers and intervention targets. However, there are multiple RNA binding sites on circRNA and miRNA that may regulate the expression of multiple target RNAs at the same time. The effects of these target RNAs may neutralize or superimpose each other on the cell phenotype, which limits the interpretation of the results in our research. Many studies have found that circRNA can bind to proteins, regulate the expression of parental genes and RNA-protein interactions at the post-transcriptional level, and participate in the regulation of various human complex diseases (such as tumors). For example, in HeLa cells of cervical cancer, the RNA binding protein HuR can promote the translation of PABPN1, and circPABPN1 (hsa_circ_0031288) can bind to HUR, which reduced the binding of HUR to PABPN1 mRNA and inhibited protein expression [32]. In addition, other studies have found that some circRNAs can recruit proteins to specific locations to play a role:for instance, the oncogene FLI1 advanced tumor metastasis not only through the classic oncoprotein pathway but also through the epigenetic mechanism mediated by recruiting demethylase TET1 [47]. Moreover, a miRNA can regulate the expression of multiple target genes, and multiple miRNAs can also regulate the expression of the same gene. The influencing factors are very complicated, so it is necessary to further explore other mechanisms. Perhaps our research is just the tip of the iceberg, but our results not only explain the potential mechanism of osteosarcoma processes regulated by circRNA but also provide developable biomarkers and therapeutic targets for osteosarcoma patients.

## Figures and Tables

**Figure 1 ijms-23-06200-f001:**
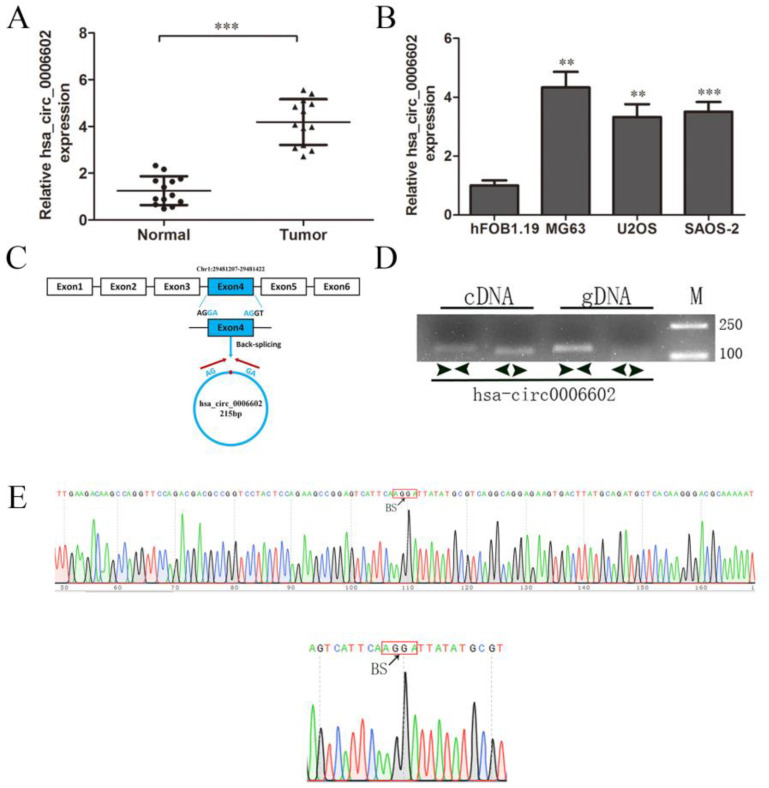
Characterization and expression of hsa_circ_0006602 in OS tissues and cell lines. (**A**) Relative expression of hsa_circ_0006602 in OS tissues and corresponding paracancerous tissues (*n* = 13). (**B**) Relative expression of hsa_circ_0006602 in OS cell lines. (**C**) Head-to-tail splicing junction in hsa_circ_0006602. (**D**) Amplification of hsa_circ_0006602 using cDNA and gDNA as templates in MG63 cells; GAPDH was used as a negative control. (**E**) Sequencing analysis of hsa_circ_0006602, ** *p* < 0.01, *** *p* < 0.001.

**Figure 2 ijms-23-06200-f002:**
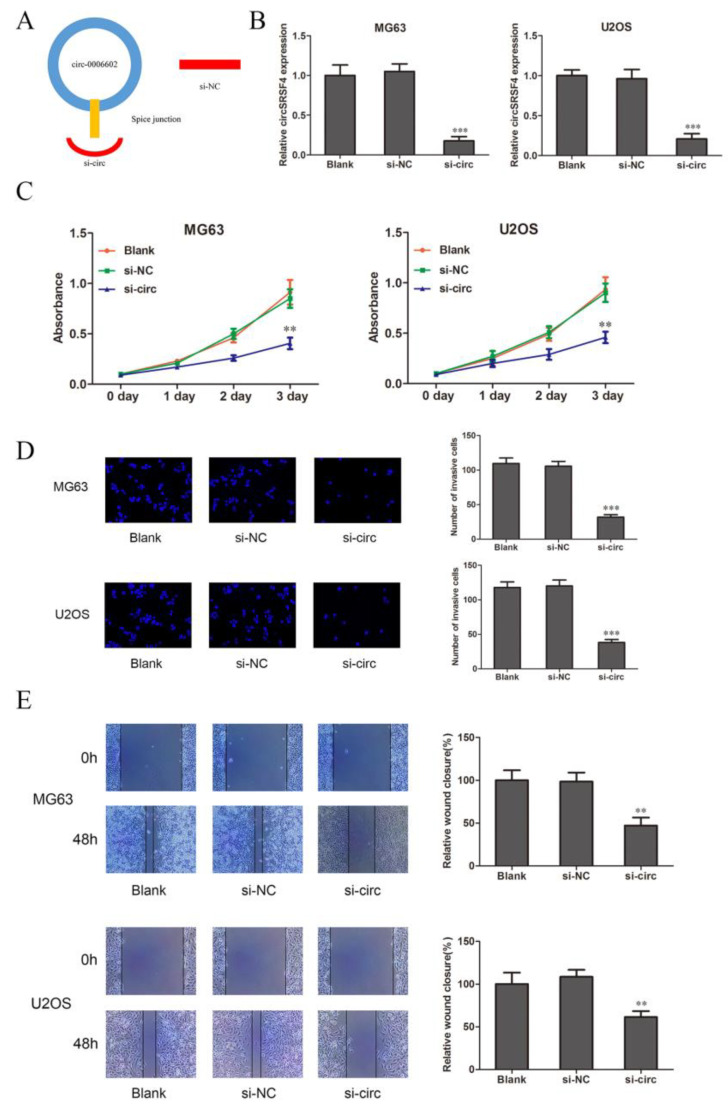
CircSRSF4 promotes the proliferation, migration, and invasion of OS cells. (**A**) Structure of circSRSF4 siRNA. (**B**) Expression of CircSRSF4 in OS cells transfected with circSRSF4 siRNA. (**C**–**E**) The proliferation activity, invasion, and migration ability of transfected cells were detected by CCK-8 assay, transwell assay, and scratch wound healing assay. ** *p* < 0.01, ****p* < 0.001.

**Figure 3 ijms-23-06200-f003:**
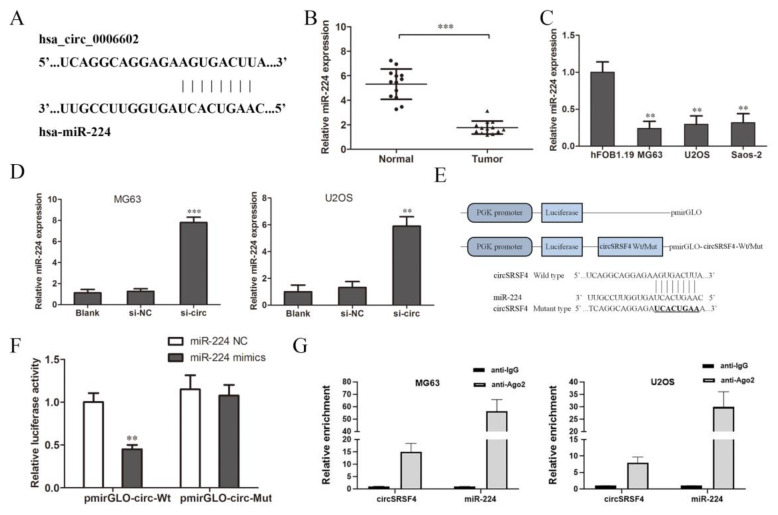
CircSRSF4 can directly bind to miR-224. (**A**) The binding site of circSRSF4 and miR-224. (**B**) Relative expression of miR-224 in OS tissues and corresponding paracancerous tissues. (**C**) Relative expression of miR-224 in OS cell lines. (**D**) Expression of miR-224 in OS cells transfected with circSRSF4 siRNA. (**E**) The structure of the pmirGLO-circSRSF4 recombinant vector. (**F**) Dual-luciferase reporter gene assay showed that circSRSF4 could bind to miR-224. (**G**) RNA binding protein immunoprecipitation (RIP) assay showed that circSRSF4 could bind to miR-224. ** *p* < 0.01, *** *p* < 0.001.

**Figure 4 ijms-23-06200-f004:**
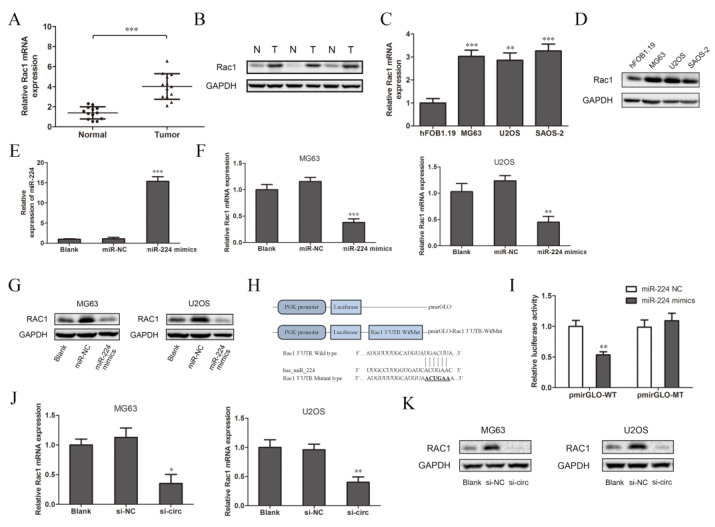
CircSRSF4 served as a ceRNA for miR-224 to regulate the expression of Rac1. (**A**) Relative expression of Rac1 mRNA in OS tissues and corresponding paracancerous tissues. (**B**) Expression of Rac1 in OS tissues and corresponding paracancerous tissues. (**C**) Relative expression of Rac1 mRNA in OS cell lines. (**D**) Expression of Rac1 in OS cell lines. (**E**) Expression of miR-224 in cells transfected with miR-224 mimics. (**F**) Relative expression of Rac1 mRNA in OS cell lines transfected with miR-224 mimics. (**G**) Expression of Rac1 in OS cell lines transfected with miR-224 mimics. (**H**) The structure of the pmirGLO-Rac1-3’UTR-WT/Mut recombinant vector. (**I**) Dual-luciferase reporter gene assay showed that miR-224 could bind to Rac1. (**J**) Relative expression of Rac1 mRNA in OS cells transfected with circSRSF4 siRNA. (**K**) Expression of Rac1 in OS cell lines transfected with circSRSF4 siRNA. * *p* < 0.05, ** *p* < 0.01, *** *p* < 0.001.

**Figure 5 ijms-23-06200-f005:**
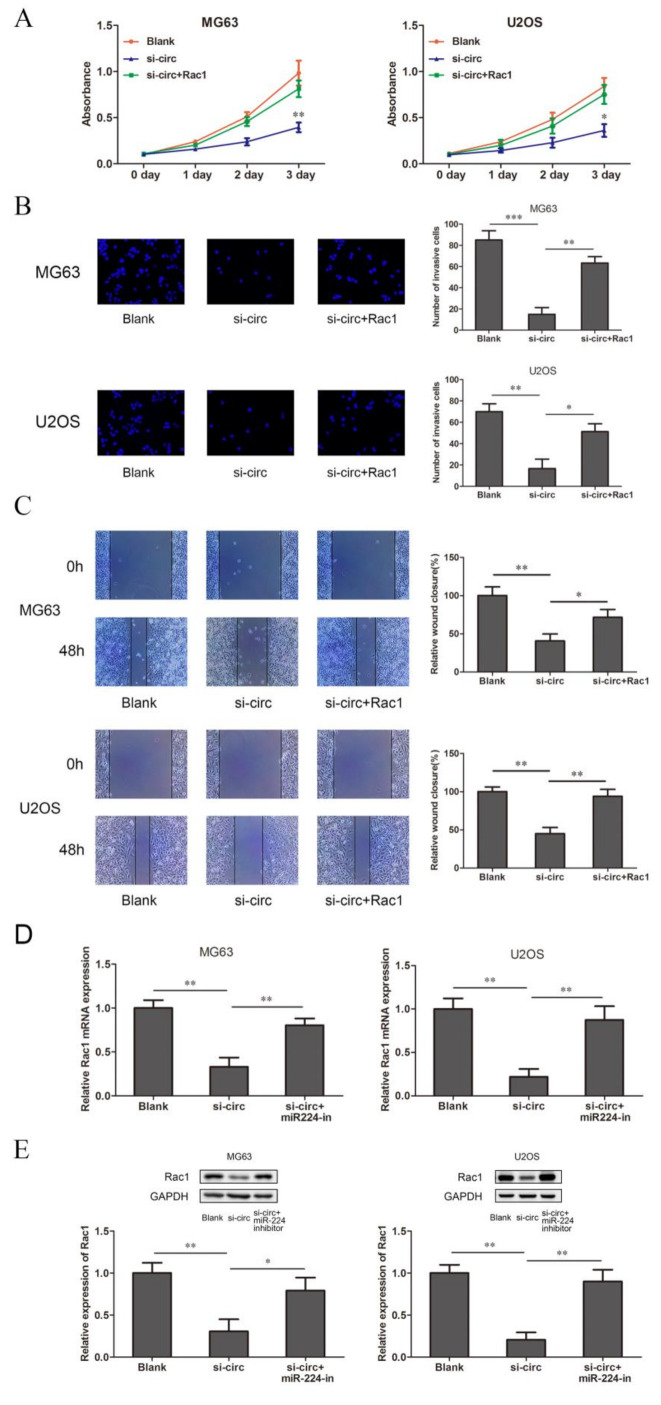
CircSRSF4 promotes OS cell proliferation, migration, and invasion through the circSRSF4/miR-224/Rac1 axis. (**A**–**C**) The proliferation activity, invasion, and migration ability of transfected cells were detected by CCK-8 assay, transwell assay, scratch wound healing assay in OS cells transfected with circSRSF4 siRNA, circSRSF4 siRNA + Rac1 respectively. (**D**,**E**) Relative mRNA and protein levels of Rac1 in OS cells transfected with circSRSF4 siRNA, circSRSF4 siRNA + miR-224 inhibitor by qRT-PCR and Western blotting, respectively. * *p* < 0.05, ** *p* < 0.01, *** *p* < 0.001.

**Figure 6 ijms-23-06200-f006:**
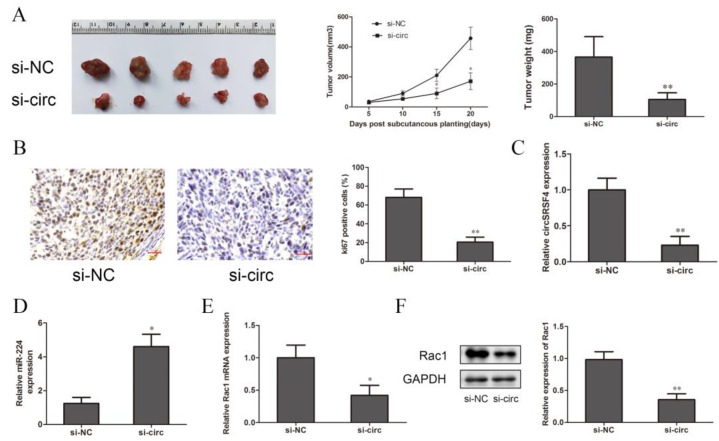
Down-regulation of circSRSF4 inhibits the growth of OS in vivo. (**A**) After knocking down circSRSF4, the growth of transplanted tumors in nude mice was inhibited. (**B**) Immunohistochemistry staining of tumor tissues in nude mice. (**C**) Expression of circSRSF4 in tumor tissues of nude mice. (**D**) Expression of miR-224 in tumor tissues of nude mice. (**E**) Expression of Rac1 mRNA in tumor tissues of nude mice. (**F**) Rac1 protein expression in tumor tissues of nude mice. * *p* < 0.05, ** *p* < 0.01.

**Table 1 ijms-23-06200-t001:** The clinicopathological features of 13 patients.

Feature	*n*
Gender Male Female	8 5
Age ≥15 <15	9 4
Tumor site Distal femur	6
Proximal femur	7
Metastasis Positive Negative	3 10

**Table 2 ijms-23-06200-t002:** The sequences of the primers.

Gene	Position	Sequence (5′ to 3′)
hsa_circ_0006602	Forward	TTGAAGACAAGCCAGGTTCC
	Reverse	ATTTTTGCGTCCCTTGTGAG
hsa-miR-224	Forward	CTGGTAGGTAAGTCACTA
	Reverse	TCAACTGGTGTCGTGGAG
U6	Forward	CAGCACATATACTAAAATTGGAACG
	Reverse	ACGAATTTGCGTGTCATCC
GAPDH	Forward	AATGGGCAGCCGTTAGGAAA
	Reverse	TGAAGGGGTCATTGATGGCA

## Data Availability

Not applicable.

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
