# Peer review of "CircSRSF4 Enhances Proliferation, Invasion, and Migration to Promote the Progression of Osteosarcoma via Rac1"

_ijms, 2022, doi:10.3390/ijms23116200_

Round 1
Reviewer 1 Report
Sun et al. nicely demonstrate that circSRSF4 expression is elevated in OS and that it contributes to OS aggressive behavior by favoring the enhanced expression of RAC1. The study presents supporting data in cell lines, patient samples and a mouse model giving significant credence to the importance of the results obtained. There are a number of points that need to be addressed.
- Almost all figures are of poor quality, and the writing in most is illegible. These must be replace for a full appreciation and interpretation of the data.
- Is the end of the paragraph in 2.2 Cell lines and cell culture, correct? Were cells other than hFOB grown at 33.5 C? If so, why?
- In section 2.8 Transwell invasion assay. As written, it appears you counted cells that remained in the upper chamber, not cells which migrated.
- In sections 2.8 and 2.9, "200x microscope" and "100x microscope" would be better stated as "200x magnification" and "100x magnification".
- As U2OS cells have been reported to be more tumorigenic and aggressive in nude mice when compared to MG63 (see Mohseny, AB et al. 2011 Lab. Invest.), why did the authors choose MG63 cells for the in vivo studies? Please make the reasoning clear in the text.
- Additionally, the authors state in section 2.13 that the MG63 cells were stably infected with lentivirus. While puromycin selection is rapid, it usually takes a minimum of 3 days for complete selection to occur. The methods state that these were selected for 24 h then used for tumor formation assays; this seems short. It also does not take into consideration the removal dead or dying cells. Please explain.
- The authors should describe their RNA immunoprecipitation protocol in the materials and methods section.
- Alternative splicing, especially of splicing factors, has been proposed as a contributing factor in the development of multiple cancers. From the results obtained in this study, do the authors think that this has more to do with circRNAs that may be being generated during alternative splicing, the formation of alternatively spliced protein isoforms, or both? Please comment on this aspect in the discussion.
- A number of minor typos/grammatical errors are present:
- Abstract: should "circsrsf4" be "circSRSF4"?
- Section 2.4: space between "amplification.Then".
- Section 2.5: space between "100ulof", "5min.At", and "uLof".
- Section 2.7: space between "uLof", "platewas", and "shaken.Then".
- Section 2.14: space between "sizeswere".
- Section 3.2: end of 1st paragraph should read, "that siRNA to circSRSF4....".
- Legend of Figure 5: rather than "circSRSF4 siRNA, and Rac1" it would be clearer if stated "circSRSF4 + Rac1".
- Section 3.6: third line, space between "volumeswere"; forth line, space between "weightswere".
- Discussion: first paragraph, second line, space between "givescircRNA"; line 16, space between "circSRSF4was".
Author Response
- Almost all figures are of poor quality, and the writing in most is illegible. These must be replace for a full appreciation and interpretation of the data.
Answer: All pictures have been corrected. We are very sorry for the mistakes in our work. The pictures were compressed during typesetting but we did not find out. Thank you very much for your valuable advice.
- Is the end of the paragraph in 2.2 Cell lines and cell culture, correct? Were cells other than hFOB grown at 33.5 ℃? If so, why?
Answer: Thank you very much for your suggestion. Sorry for the mistake, and it has been corrected. Only hFOB 1.19 cells were cultured at 33.5 ℃, and other cell lines were cultured at 37 ℃. The hFOB 1.19 cell line is very special and proliferation was seen only for cells grown at 33.5 ℃. It was developed by conditionally immortalizing human fetal osteoblasts with a temperature-sensitive mutant of the SV40 large T antigen (ts-SV40LTA) gene. At the permissive temperature of 33.5 ℃, the ts-SV40LTA is active and the hFOB cells proliferate rapidly (see MEN-LUH YEN et al. 2007, PMID: 17204605).
- In section 2.8 Transwell invasion assay. As written, it appears you counted cells that remained in the upper chamber, not cells which migrated.
Answer: Thanks for your advice. The description was inappropriate and now it has been changed: After gently wiping the upper chamber with a cotton swab, we placed the chamber in 4% paraformaldehyde for 15 minutes, washed it three times with PBS, and then stained it with DAPI for 5 minutes.
- In sections 2.8 and 2.9, "200x microscope" and "100x microscope" would be better stated as "200x magnification" and "100x magnification".
Answer: Your proposal has been shown in the manuscript and thank you very much for your suggestion.
- As U2OS cells have been reported to be more tumorigenic and aggressive in nude mice when compared to MG63 (see Mohseny, AB et al. 2011 Lab. Invest.), why did the authors choose MG63 cells for the in vivo studies? Please make the reasoning clear in the text.
Answer: Thank you very much for your advice. Actually, it has been found in previous studies that both MG63 and U2OS cells can be used for xenograft model in nude mice (see Masanori Kawano et al. 2021; Jinyu Zhang et al. 2020). Our team used MG63 for many times and modeled successfully. For the sake of success, we also selected MG63 cells.
- Additionally, the authors state in section 2.13 that the MG63 cells were stably infected with lentivirus. While puromycin selection is rapid, it usually takes a minimum of 3 days for complete selection to occur. The methods state that these were selected for 24 h then used for tumor formation assays; this seems short. It also does not take into consideration the removal dead or dying cells. Please explain.
Answer: Thank you very much for your comments. Some descriptions are missing there and have been added now. After culturing for 24 hours, the medium was replaced with fresh medium, and the optimal concentration of puromycin was added after 48 hours to screen the cells. After 72 hours, the surviving infected cells were collected and used for a tumor formation experiment in nude mice.
- The authors should describe their RNA immunoprecipitation protocol in the materials and methods section.
Answer: According to the protocol of EZ-Magna RIP kit (Millipore), after collecting 107 cells, RIP lysis buffer containing protease inhibitor cocktail and RNase inhibitor was added, followed by 40 minutes on ice and centrifuged at 4 ℃ for 15 minutes (12000g). Then incubate 4 hours at 4 ℃ with RIP buffer containing magnetic beads bound to anti-argonaute 2 (Ago2) antibodies and negative control IgG. After RNA extraction from the samples, the enrichment of circSRSF4 was identified by q-PCR. The whole experiment should be performed under the condition of RNase free.
- Alternative splicing, especially of splicing factors, has been proposed as a contributing factor in the development of multiple cancers. From the results obtained in this study, do the authors think that this has more to do with circRNAs that may be being generated during alternative splicing, the formation of alternatively spliced protein isoforms, or both? Please comment on this aspect in the discussion.
Answer: Thank you for the proposal, which can enrich the discussion: Nigro et al. first discovered transcripts produced by non-canonical splicing caused by deleted in colon cancer gene. Later, Cocquerelle proved the circularization of these transcripts, but he called these molecules 'mis-spliced'. Although the function of circRNAs has been discovered gradually with the deepening of research, there is still a opinion that they could be a by-product of splicing. Until 2014, someone demonstrated that circRNAs are produced co-transcriptionally, moreover, they could compete with conventional splicing. Therefore, the production of circRNA results in decreased mRNA at the same locus. CircRNA is produced by spliceosomes, however, there are many circular RNAs formed by exons that are not produced by alternative splicing. Thus, some enrichment of circRNAs could regulate the production of host gene mRNAs. In addition to alternative splicing, the production of circRNAs is also associated with inefficient cleavage and polyadenylation.
- A number of minor typos/grammatical errors are present.
Answer: Thanks for your suggestion and the errors has been revised.
Reviewer 2 Report
This is an interesting manuscript demonstrating the biological role of circular RNAs is osteosarcoma development. This is a phenomenon described only recently, however, it may play significant role in carcinogenesis. the authors show interactions of cirSRSF4 with Rac1 through miR-224 and based on their results conclude that cirSRSF4 may be a new putative therapeutic target in osteosarcoma.
The study is adequately structures and described including necessary ethical standards concerning its patient part. Methods are duly described. Animal part of the methods is also concisely described with necessary standards. The results are clearly presented with comprehensive description of the respective parts of this study.
The list of references appears adequate and comprehensive. The manuscript is written in good English language.
Author Response
Thank you very much for your positive evaluation of our study, we will continue to improve the content.
Round 2
Reviewer 1 Report
The authors have responded adequately to my comments. The figures are a big improvement. Some minor proofing would assist in correcting the few grammatical errors present.
The authors should fix the beginning of the sentence, "Until 2014, someone demonstrated..."
Once these
This manuscript is a resubmission of an earlier submission. The following is a list of the peer review reports and author responses from that submission.